# Peer review of "Recent Advances in Progresses and Prospects of IL-37 in Central Nervous System Diseases"

_brainsci, 2022, doi:10.3390/brainsci12060723_

Round 1
Reviewer 1 Report
No suggestions
Author Response
Thank you very much for your review.
Reviewer 2 Report
The study by Li et al. ‘Recent Advances in Progresses and Prospects of IL-37 in Central Nervous System Diseases’, is interesting and has significance to the field of CNS such as cute spinal cord injury (ASCI), Demyelinating disease, Alzheimer's diseases, and Stroke.
The authors summarized the existing data on the function of IL-37 in CNS diseases. However, since the function of IL-37 in AD has not been elucidated yet, the authors could not add much information, but the authors suggested a direction for future research. There are few major and minor comments are:
Major comments,
The authors seem confused, but this is a very important part.
- Figure1, IL-18a needs to be changed to IL-18Ra in the cartoon. Also, authors should change IL 37/IL-18a/IL1R8 to IL 37/IL-18Ra/IL1R8.
- Authors should change IL-18a chain to IL-18Ra chain in the Line 302.
Minor comments.
- The author confuses the symbols of genes and proteins. Therefore, please write the symbol of the gene again according to the example below. For example, IL-37 in the Line 19 is protein and it is correct but IL-37 in Line 22 needs to change IL-37 with italic letters. And according to the reference, they use human IL-37, so you can use all uppercase letters.
Suggested Guidelines: Humans, non-human primates, chickens, and domestic species: Gene symbols contain three to six italicized characters that are all in upper-case (e.g., AFP). Gene symbols may be a combination of letters and Arabic numerals (e.g., 1, 2, 3), but should always begin with a letter; they generally do not contain Roman numerals (e.g., I, II, III), Greek letters (e.g., α, β, γ), or punctuation. Protein symbols are identical to their corresponding gene symbols except that they are not italicized (e.g., AFP). Mice and rats: Gene symbols are italicized, with only the first letter in upper-case (e.g., Gfap). Protein symbols are not italicized, and all letters are in upper-case (e.g., GFAP).
- Line 83: Use the full word of LPS and use the symbol such as Lipopolysaccharides (LPS).
- Authors should change micromolar to picomolar in the Line 286.
- Please re-write sentence between line 22-26.
- Please add answers to above questions and requests to your manuscript.

Reviewer 3 Report
well structured manuscript, scientifically vlid. the date are correct, thr conclusions are consistent.
Author Response
Thank you for your review and affirmation.